# MiR-138-5p Upregulation during Neuronal Maturation Parallels with an Increase in Neuronal Survival

**DOI:** 10.3390/ijms242216509

**Published:** 2023-11-20

**Authors:** María Asunción Barreda-Manso, Altea Soto, Teresa Muñoz-Galdeano, David Reigada, Manuel Nieto-Díaz, Rodrigo M. Maza

**Affiliations:** 1Research Unit, Molecular Neuroprotection Group, Hospital Nacional de Parapléjicos, SESCAM, 45071 Toledo, Spain; alteas@sescam.jccm.es (A.S.); tmunozd@sescam.jccm.es (T.M.-G.); dreigada@sescam.jccm.es (D.R.); mnietod@sescam.jccm.es (M.N.-D.); 2Research Unit, Functional Exploration and Neuromodulation of the Central Nervous System (FENNSI) Group, Hospital Nacional de Parapléjicos, SESCAM, 45071 Toledo, Spain

**Keywords:** miR-138-5p, maturation, neuroprotection, apoptosis, neuronal death

## Abstract

Neuronal maturation is a process that plays a key role in the development and regeneration of the central nervous system. Although embryonic brain development and neurodegeneration have received considerable attention, the events that govern postnatal neuronal maturation are less understood. Among the mechanisms influencing such neuronal maturation processes, apoptosis plays a key role. Several regulators have been described to modulate apoptosis, including post-transcriptional regulation by microRNAs. This study aimed to analyze endogenous expression changes of miR-138-5p, as well as its main validated pro-apoptotic target caspase3, during the maturation of neuronal cultures and their response under apoptotic challenge. Our results point out that the observed opposite expression of miR-138-5p and its target caspase3 might modulate apoptosis favoring neuronal survival at distinct maturation stages. The unchanged expression of miR-138-5p in mature neurons contrasts with the significant downregulation in immature neurons upon apoptotic stimulation. Similarly, immunoblot and individual cellular assays confirmed that during maturation, not only the expression but processing of CASP-3 and caspase activity is reduced after apoptotic stimulation which results in a reduction of neuronal death. Further studies would be needed to determine a more detailed role of miR-138-5p in apoptosis during neuronal maturation and the synergistic action of several microRNAs acting cooperatively on caspase3 or other apoptotic targets.

## 1. Introduction

Neurogenesis is the process of the formation of new neurons and involves the proliferation, migration, differentiation, and integration of neurons into the existing neuronal circuit. Within these four neurogenesis phases, neuronal maturation plays an essential role, which is not only relevant during the development of the individual but also in the creation of new neurons in the adult individual for essential processes such as memory [1] or even during neurodegenerative diseases or injury repair [2]. Because of this, neuronal maturation has a strong impact on the understanding of the functioning of the central nervous system (CNS) in both physiological and pathological conditions. Therefore, it is essential to study the mechanisms of action underlying this maturation in order to promote it and, thus, achieve the necessary regeneration in many pathological and traumatic processes [3].

Neuronal maturation is a complex process in which, over days and even weeks, neurons change morphologically, including axonal growth, dendrite formation, and connection of the dendritic tree with neighboring neurons [4]. Various factors cause these changes, such as environment [5,6], growth factors (e.g., NGF) [7,8,9], and transcriptome [10,11]. The modulation of transcriptome by non-coding RNA includes microRNAs, small RNA molecules of approximately 19–25 nucleotides capable of controlling the expression of hundreds of genes and regulating the state and fate of cells [12]. MicroRNAs have a pivotal role throughout neuronal development and maturation, from the formation of different neuronal types to the decision of neuronal plasticity and connections with other neurons [13,14,15,16,17].

Another important factor in the neuroregeneration of the pathological nervous system is that approximately 50% of newborn neurons die because of the execution of programmed cell death [18], even though during both developmental and adult neurogenesis the number of neurons that will die or mature varies considerably depending on the post-differentiation time, neuronal type, and CNS region [19]. Among all of the mechanisms that prevent new neurons from maturing is the direction of them toward apoptotic death [20,21]. The employment of multiple and redundant mechanisms to inhibit apoptosis enables mature neurons, with long-term survival capabilities, to become strikingly resistant to injury after the developmental stage [22,23]. Indeed, maturation of the CNS is associated with a decrease in the expression of several caspase genes [24], including the effector caspase-3/7 [22].

Post-transcriptional regulation by microRNAs has been described among many modulators of apoptotic effectors (such as caspases) during the maturation of neuronal cells. Several microRNAs target the Casp3 gene (coding for caspase-3 or CASP-3 protein) and control its expression, such as miR-138-5p [25,26]. This microRNA is one of the most enriched in the CNS [27,28], mainly expressed in both brain and spinal cord neurons and varying depending on the neuronal subtype [26]. It has a heterogeneous expression and distribution throughout the neural tissue, which changes during the CNS’s development [27,29,30]. Moreover, it has also been shown that its expression levels change over the days of neuronal culture in vitro [31]. On the other hand, miR-138-5p is known to regulate apoptotic cell death by regulating the expression of pro-apoptotic genes, among which include Casp3 [26]. In the present work, we hypothesize that changes in miR-138-5p expression during neuronal maturation could control pro-apoptotic caspase 3 expression and activity, thereby aiding in the survival of neurons exposed to noxious stimuli.

## 2. Results

### 2.1. Opposite Expression of miR-138-5p and Caspase 3 during Hippocampal Neuronal Maturation

In order to establish miR-138-5p expression during hippocampal neuronal maturation and to evaluate its possible effect on apoptotic proteins, we studied miR-138-5p levels in hippocampal neuron cultures over 18 days of maturation. RT-qPCR analyses show a significant upregulation of miR-138-5p at 7, 14, and 18 days of culture related to 1 day (*p* < 0.001, see Figure 1A and the table in Appendix A for gene expression details; observe that ΔCt values are inversely related to gene expression). In the case of the expression of its target apoptotic gene, caspase 3, we observed a significant decrease in Casp3 levels from 14 days of culture (*p* < 0.05, see Figure 1B and the table in Appendix A). In addition, as shown in Figure 1C, D, immunoblot assays confirm the significant downregulation of pro-CASP-3 expression at 14 days of culture related to 1 day (14 d = 0.24% ± 0.07 vs. 1 d = 0.85% ± 0.11; two-tailed paired *t*-test, T_3_ = 13.21, *p* = 0.0009).

These results indicate that the changes in the expression of miR-138-5p and caspase 3 were opposite after 14 days, so for the following experiments related to neuroprotection we selected cells at 1 and 14 days of maturation.

### 2.2. Neuronal Purity of Hippocampal Cell Cultures

Before performing the different experiments with the cytotoxic treatment, we measured the purity of 1 and 14 days hippocampal neuronal cultures to avoid biases and correctly assign the neuronal effect in the following functional analysis. We observed that the percentage of neurons (β−III-tubulin-stained cells) at 1 day of culture related to total cells in culture was 82.02% ± 10.76. However, at 14 days this percentage decreased to 66.69% ± 11.62 (Figure 2). To avoid making a neuronal counting error due to this difference in purity over the days of culture, we only considered cells with a neuronal phenotype in the remaining assays performed in the single-cell analysis.

### 2.3. Neuronal Maturation Attenuates Caspase-Dependent Apoptosis

In order to evaluate the injury response throughout the days of maturation, we analyze the cellular levels of caspase 3 in the hippocampal neurons stimulated or not with L-glutamic acid (LGA). We observed that treatment with LGA did not change the endogenous protein levels of pro-CASP-3 on neurons relative to untreated cells (Ctr) neither at 1 day of culture (LGA = 87.40% ± 17.87 vs. Ctr (unstimulated after 1 day culture) = 100%; Figure 3A,B) nor at 14 days of maturation (LGA = 111.16% ± 16.64 vs. Ctr (unstimulated after 14 days culture) = 100%; Figure 3A,B).

A comparison of the protein expression levels of cleaved-CASP-3 in hippocampal neurons stimulated with LGA reveals that cleaved-CASP-3 expression did not change at 1 day of culture (LGA = 103.39% ± 36.95 vs. Ctr (unstimulated after 1 day culture) = 100%; Figure 3A,C). However, at 14 days of maturation, the expression levels of cleaved-CASP-3 in neuronal cultures stimulated with LGA decreased related to unstimulated neurons (LGA = 78.49% ± 30.83 vs. Ctr (unstimulated after 14 days culture) = 100%; Figure 3A,C). Thus, a significant change in the ratio of cleaved-CASP-3 levels induced by LGA in neurons at 14 days relative to the ratio of cleaved-CASP-3 levels induced by LGA in neurons at 1 day was observed (two-tailed paired *t*-test, T_2_ = 5.461, *p* = 0.032; Figure 3C).

### 2.4. Neuronal Maturation Reduces Caspase Activity

In order to gain insight into the involvement of neuronal active effector caspase, we quantified the enzymatic activity of effector caspase-3/7 after LGA treatment in single hippocampal neurons. As shown in Figure 4A, at both 1 and 14 days, the activity of the effector caspase-3/7 of LGA-treated cells increased related to the unstimulated neurons (Ctr) (1 d: LGA = 204.4% ± 71.68 vs. Ctr (unstimulated after 1 day culture) = 100%; 14 d: LGA = 120.3% ± 48.32 vs. Ctr (unstimulated after 14 days culture) = 100%). We also tested whether maturation also reduced the caspase-3/7 activity in the cultures of hippocampal neurons after cytotoxic stimulation with LGA. The increase in the enzymatic activity of the neuronal effector caspase-3/7 induced by LGA was significantly lower in neurons at 14 days of culture than at 1 day (1 d = 1.04 ± 0.72, 14 d = 0.20 ± 0.48; two-tailed paired *t*-test, T_5_ = 4.265, *p* = 0.008; Figure 4B).

Despite observing different cell numbers at 1 and 14 days of culture, we did not observe differences in the neuronal density (neurons per mm^2^) under the different culture conditions (maturation days or treatment) (1 d Ctr = 159.55 ± 103.76, LGA = 161.96 ± 171.67; 14 d Ctr = 192.06 ± 77.96, LGA = 143.54 ± 90.41) (Figure 4C).

### 2.5. Cell Maturation Promotes Neuronal Survival

To evaluate whether the changes in expression of cleaved-CASP-3 and caspase activity affect neuronal survival, we stimulated hippocampal neuron cultures with LGA at 1 and 14 days of culture and evaluated the percentage of live and dying cells in each sample. As shown in Figure 5, at both 1 and 14 days, when we treated cells with LGA, the neuronal survival (calcein-AM-stained cells) was reduced (1 d: LGA = 81.86% ± 6.75 vs. Ctr (unstimulated after 1 day culture) = 100%; 14 d: LGA = 95.60% ± 10.43 vs. Ctr (unstimulated after 14 days culture) = 100%) and cell death (PI-stained cells) increased (1 d: LGA = 209.2% ± 50.83 vs. Ctr (unstimulated after 1 day culture) = 100%; 14 d: LGA = 113.3% ± 48.22 vs. Ctr (unstimulated after 14 days culture) = 100%). On the other hand, and as shown in Figure 5A,B, the decrease in neuronal survival induced by stimulation with LGA was significantly lower in neurons at 14 days of culture than at 1 day (1 d = −0.18 ± 0.07, 14 d = −0.06 ± 0.10; two-tailed paired *t*-test, T_3_ = 5.321, *p* = 0.007). We confirmed the difference in cell death over the days of maturation using a propidium iodide (PI) assay. Complementary, hippocampal neuron cultures treated with LGA significantly reduced cell death at 14 days related to 1 day of culture (1 d = 1.09 ± 0.51, 14 d = 0.13 ± 0.48; two-tailed paired *t*-test, T_4_ = 4.227, *p* = 0.013; see Figure 5A,C).

As in the caspase activity assay, we observed different numbers of total cells at 1 and 14 days of culture. However, we did not observe any differences in the neuronal density in these cell survival or death assays under the different culture conditions (1 d Ctr = 157.28 ± 123.33, LGA = 160.95 ± 109.75; 14 d Ctr = 184.94 ± 85.63, LGA = 127.94 ± 51.76) (Figure 5D).

Altogether, these results indicate that cell maturation promotes neuronal survival in cytotoxic conditions.

### 2.6. Neuronal Maturation Does Not Change the Increase in miR-138-5p Expression in Apoptotic-Stimulated Hippocampal Neurons

To evaluate whether the levels of the Casp3-targeted miR-138-5p change under cytotoxic conditions and to establish a possible relationship with apoptosis, we stimulated hippocampal neuron cultures with LGA at 1 and 14 days of culture and studied the endogenous expression of miR-138-5p. Initially, RT-qPCR analyses confirmed the upregulation of miR-138-5p at 14 days related to 1 day of culture in endogenous conditions (Ctr) (14 d = 1.87 ± 0.29 vs. 1 d = 2.57 ± 0.67; observe that ΔCt values are inversely related to gene expression), as well as after LGA treatment (14 d = 1.85 ± 0.11 vs. 1 d = 2.81 ± 0.81). In addition, these data show the marginally significant downregulation of miR-138-5p after LGA stimulation at 1 day of culture (two-tailed paired *t*-test, T_3_ = 2.854, *p* = 0.1; Figure 6 and the table in Appendix A). However, at 14 days of maturation, the expression of miR-138-5p after LGA treatment did not change related to the unstimulated neurons (Ctr; Figure 6 and the table in Appendix A). With these results, we proved that the increase in miR-138-5p expression throughout maturation does not change under apoptotic stimulation.

## 3. Discussion

CNS development depends on a sculpting process that removes neural cells through programmed cell death. Apoptosis is widely acknowledged as the main regulator of the number of cells during development, which needs to be controlled in order to avoid neuronal overload and deleterious malformations in the nervous system [32,33,34,35]. On the other hand, the modulation of apoptotic activation in newborn neurons in pathological situations, such as injury or neurodegenerative diseases, is essential for neuronal maturation and tissue regeneration [36]. Thus, the regulation of neuronal maturation is key to understanding not only how neurons survive over the life of an organism but sheds light on how neuronal injury in neurodegenerative pathologies can lead to cell death. Among the numerous factors involved in determining cell fate during CNS maturation (e.g., cellular environment, growth factors, and transcriptional regulation) [37], we focused on the microRNA post-transcriptional regulation of pro-apoptotic factors. In this study, we observed an opposite pattern in the change in the expression of endogenous Casp3 and miR-138-5p during hippocampal neuron maturation. Interestingly, miR-138-5p expression in mature neurons is not altered following apoptotic stimulation in contrast to the significant downregulation in immature neurons. Similarly, during neuronal maturation, not only the expression but also the processing of CASP-3 and caspase activity are reduced after apoptotic stimulation, resulting in a decrease in cell death.

Our results showing a decrease in Casp3 expression during maturation agrees with other studies that show that the development of the nervous system is associated with the decreased expression of caspase 3 [38] or other pro-apoptotic effectors, such as caspase 7 or BAK [23,39]. These changes are due, among other factors, to changes in their transcriptional regulation during neuronal development or maturation [23,40]. The rate of transcription initiation of Casp3 substantially declines during brain maturation and is associated with transcriptional silencing by epigenetic regulation, such as differential DNA methylation and histone acetylation [41]. In addition to transcriptional regulation, post-transcriptional control of gene expression by microRNAs is crucial for the different phases of neuronal maturation [16]. Several microRNAs are key regulators of neuronal maturation, acting as inhibitors of neuronal apoptosis, such as miR-29, which modulates Bim, Bmf, and Puma protein expression [42,43]. Similarly, miR-138-5p, which is involved in migration, axonal growth, or in determining the size of dendritic spines [44,45,46], has been shown to regulate both the expression and activity of pro-apoptotic factors (i.e., Casp3, Casp7, and Bak1) following traumatic spinal cord injury [26]. Thus, microRNAs are integrated into a broader regulatory network that governs the dynamic processes of CNS maturation, such as neuronal apoptosis, neuronal differentiation, and neuronal proliferation [16].

There are multiple factors and molecular mechanisms that both transcriptionally and post-transcriptionally regulate microRNA expression and that may influence the CNS developmental processes. At the transcriptional level, microRNAs undergo regulation by transcription factors associated with neuronal maturation and survival. Thus, the nuclear TLX receptor, known for its involvement in neuronal progenitor self-renewal, exerts a regulatory influence on miR-9 expression levels, playing a key role in controlling the balance between neural stem cell renewal and differentiation [47]. Another example is miR-138-5p, which has been described to be transcriptionally upregulated during myelination and downregulated upon nerve injury, but not much is known about the factors that might produce its downregulation during CNS development [48]. A second level of regulation is the post-transcriptional control of microRNAs, which is based on microRNA processing, stability, and repressive activity during neuronal development. Precursors of mature microRNAs are subject to regulation, such as the inhibition of miR-547 by amyloid precursor protein (APP) in the developing cerebral cortex through the induction of pri-miR-547 degradation or the degradation of the pro-neural microRNA miR-9 by Lin28 at the pre-microRNA level [49,50]. Interestingly, a differential processing of precursor of miR-138-5p into mature microRNAs has been reported to lead to tissue- and developmental-specific microRNA expression in mammals [27]. On the other hand, different RNA classes with microRNA-repressive functions can regulate microRNA activity. Competing for endogenous RNAs (ce-RNAs), long non-coding RNAs (e.g., LncNDs), or circular RNAs (circRNAs), such as ciRS-7, can function as microRNA sponges [16]. Finally, the stability of microRNAs can be regulated by modifications (e.g., methylation and adenylation) or by specific interaction with ribonucleases or proteins to shorten or prolong their half-life [51]. The precise impact of transcriptional and post-transcriptional regulatory mechanisms on the functionality of miR-138-5p in modulating the apoptotic processes within developing and mature neurons remains unknown. Further research is needed to unravel the regulatory networks governing miR-138-5p and its involvement in apoptotic pathways across neuronal developmental stages. This microRNA is upregulated during CNS maturation, as observed previously in cortical development [45]. From our results, we observed an increased expression level of miR-138-5p in cultured hippocampal neurons during maturation, described before in cortical neuron cultures by Weiss and colleagues [31]. These expression changes during cellular maturation are not restricted only to neurons but also occur in other cell types, being upregulated in oligodendrocytes, promoting the early stages of maturation and facilitating the appropriated axonal myelination [52]. Thus, although maturation is promoted in both cases, the function of miR-138-5p’s overexpression could be cell-dependent. In summary, our results may suggest that the regulation of apoptosis during CNS development involves a dynamic interplay of transcriptional and post-transcriptional mechanisms. Notably, this regulatory landscape encompasses microRNA-mediated control, with miR-138-5p emerging as a potential key player in modulating the expression and activity of caspase 3, thereby influencing the neuronal survival throughout CNS maturation.

Interestingly, upon exposure to a challenging cellular environment, the expression of these post-transcriptional regulators is also differentially altered depending on the neural maturation stage. The expression of miR-29a or miR-124 is altered during the maturation of neural cells in noxious conditions [53,54]. The Casp3 regulator miR-29b is downregulated in immature cerebellar neurons after ethanol exposure and recovers its high expression levels during the following days of culture, thus protecting cells from apoptotic activation [55]. Similarly, we have shown that the expression of miR-138-5p in mature neurons is not altered under cytotoxic conditions (LGA stimulus), though a marginally significant difference in downregulation was observed in young neurons. So, our results might indicate that the distinct neuronal death sensibility according to its stage of maturation might depend on the altered expression of miR-138-5p and many other microRNAs.

Our results on neuronal maturation showed the attenuation of caspase-dependent apoptosis and the promotion of neuronal survival. We observed, studying hippocampal neurons individually, that the number of neurons with effector caspase-3/7 activity decreased significantly under cytotoxic conditions (LGA stimulus) in mature neurons. Similarly, age-dependent differences have been shown in both injury-induced caspase 3 activation and susceptibility to apoptosis in the mammalian brain [56,57,58], as well as a reduction of etoposide-induced apoptosis and decreased levels of caspase 3 activity in primary rat cortical neurons [59]. Caspase-3/7 effector activity has different levels of regulation at the transcription, post-transcription, and translation stages. To post-transcriptionally control the expression of the pro-apoptotic factors, such as Casp3, during the different phases of neuronal maturation, microRNA regulation, differential subcellular compartmentalization of specific effector caspases, and changes in the expression of direct regulators of caspase 3 activity have been considered. The spatial confinement of active caspase 3, in which the control of its activity may be different in the soma than in dendrites [60], has been described, since it is also involved in axon guidance, synapse formation, axon pruning, and synaptic functions aside from their roles in eliminating unnecessary neural cells [35]. On the other hand, the attenuation of caspase 3 activation during maturation may result from the repression of factors involved in the caspase 3 activation pathway, as the observed reduction of cytochrome c’s ability to induce the activation of Casp3 in the murine brain is undetectable after 2 weeks of age [59]. However, other regulators, such as the X-linked inhibitor of apoptosis protein (XIAP), have been shown to decrease their levels in the cerebral cortex both in aging and in cultured cortical neurons [61].

## 4. Materials and Methods

### 4.1. Cell Culture

We prepared primary cultures of hippocampal neurons from 18-day-old (E18) Wistar rat embryos (RRID: RGD_13508588). After dissection from the brain, we dissociated hippocampi by incubation with 1x trypsin (ThermoFisher, Waltham, MA, USA) in Hanks’ Balanced Salt Solution (HBSS) medium without calcium and magnesium (Hyclone, Washington, DC, USA; GE Healthcare, Chigaco, IL, USA) supplemented with 20 mg/mL DNase (Roche) for 15 min at 37 °C. We washed-out trypsin solution with HBSS with calcium and magnesium (Hyclone) before dissociating the tissue through repeated pipetting in Minimum Essential Medium (MEM; Gibco, Waltham, MA, USA) supplemented with 10% horse serum (Fisher Scientific). We seeded the so-obtained cell suspension in 10 µg/mL poly-L-lysine (Sigma-Aldrich, St. Louis, MO, USA) precoated plates and let the cells adhere for 4 h at 37 °C and 5% CO_2_ in a cell culture incubator. Afterwards, we changed the medium to Neurobasal Medium (Gibco) enriched with 2% B-27 supplement (Gibco), 1% GlutaMAX (Gibco), and 100 µ/mL penicillin/streptomycin (Gibco) and kept the culture in a humidified incubator in an atmosphere of 5% CO_2_ at 37 °C for 1 or 14 days before subsequent experimental procedures.

### 4.2. Immunofluorescence Assay

We seeded 35,000 hippocampal neurons per well in 24-well plates pre-coated with 10 µg/mL poly-L-lysine. One or 14 days later, we fixed cells with 4% PFA for 30 min and washed them with PBS 1x. Then, we permeabilized and blocked neurons in blocking buffer (3% BSA (Sigma-Aldrich, St. Louis, MO, USA) and 0.2% Triton X-100 diluted in PBS 1x buffer) for 1 h at RT and incubated with antibody against β-III-tubulin (mouse anti-β−III-tubulin isoform, 1:500; Millipore cat#MAB1637, RRID: AB_2210524) and antibody against GFAP (chicken anti-GFAP, 1:1000; Abcam cat#ab4674, RRID: AB_304558) overnight (O/N) at 4 °C. Antibodies were functionally validated by Millipore and Abcam companies. After incubation with primary antibodies, we washed cells with PBS 1x and incubated them for 1 h at room temperature (RT) with a fluorescent Alexa 488-conjugated goat anti-mouse IgG secondary antibody (1:500; Molecular Probes cat#A-11029, RRID: AB_2534088) and a fluorescent Alexa 594-conjugated goat anti-chicken IgG secondary antibody (1:500; Molecular Probes cat#A-11042, RRID: AB_2534099). Finally, after three washes with PBS 1x, we mounted the coverslips on glass slides employing Fluorescence Mounting Medium (Thermo Scientific) with 1:30,000 of the fluorescent marker of nucleic acids 4’,6-diamino-2-fenilindol (DAPI; Sigma-Aldrich, St. Louis, MO, USA). We took photographs of the cells using an epifluorescence microscope (DMIL LED, Leica Microsystem GmbH, Wetzlar, Germany) with a 20× microscope lens, coupled to a Leica DFC 3000G camera. We used the ImageJ software v.1.53c (National Institutes of Health, NIH, Bethesda, MD, USA) [62] to process and analyze the images.

We quantified the purity of the hippocampal neuronal culture at 1 or 14 days of culture from the percentage of β-III-tubulin-stained neurons related to the total number of cells stained with DAPI (we analyzed a total of nine images per condition). On the other hand, we estimated the hippocampal neuronal density by calculating the total number of neurons per mm^2^ in the different cultures (we analyzed a total of five images of 0.27 mm^2^ per condition).

### 4.3. RT-qPCR Analysis

We seeded 500,000 hippocampal neurons per well in 12-well plates with coverslips pre-coated with 50 µg/mL poly-L-lysine. Next, 1, 4, 7, 14, or 18 days later, we isolated and purified total RNA from neurons with miRNeasy Kit (Qiagen, Hilden, Germany). For stimulated assays, we treated 1 or 14 days primary cultures of hippocampal neurons O/N with 15 mM LGA (Sigma-Aldrich, St. Louis, MO, USA). The total RNA concentration and purity (260/280 and 260/230 ratios) were estimated with a NanoDrop ND-1000 spectrophotometer (Thermo Scientific). Only samples with 260/280 ratios between 1.8 and 2.2 were employed.

To determine the miR-138-5p expression, 10 ng of total RNA was reverse-transcribed and amplified using a TaqMan microRNA gene expression assay (TaqMan^®^ MicroRNA assay cat#002284, Applied Biosystems, Waltham, MA, USA) following the manufacturer’s protocols. The U6 small nuclear RNA served as an internal control (TaqMan^®^ MicroRNA assay cat#001973, Applied Biosystems). For the mRNA detection of Casp3 transcripts, 1 µg of total RNA was subjected to random reverse transcription using Moloney Murine Leukemia Virus reverse transcriptase (M-MLV-RT; Invitrogen, Waltham, MA, USA) and random primers (Roche, Basel, Switzerland). Then, we evaluated the gene expression levels using TaqMan Gene Expression Assays for Casp3 (TaqMan^®^ Gene Expression Assays cat#00563962; Applied Biosystems), employing 18S ribosomal RNA (TaqMan^®^ Gene Expression Assays cat#4333760; Applied Biosystems) as a housekeeping gene. We measured the abundance of miR-138-5p and the mRNAs of interest in a thermocycler ABI Prism 7900 fast (Applied Biosystems) for 40 cycles of two steps: 15 s at 95 °C plus 1 min at 60 °C using the 2^−ΔΔCt^ method [63]. Briefly, the difference (ΔCt) between the cycle threshold of the microRNA or the target mRNA and their respective endogenous loading controls (U6 for miR-138-5p and 18S for Casp3) was estimated together with its associated variance following the standard propagation of error method from Headrick [64]. Then, we compared the ΔCt value from different times of maturation with the ΔCt from 1 day to calculate the ΔΔCt and the correspondent fold increase (2^−ΔΔCt^), indicating also the 95% confidence interval (CI). In stimulated assays, we compared the ΔCt value from treated cells with the ΔCt from unstimulated cells (Ctr) to calculate the ΔΔCt and the correspondent fold change (2^−ΔΔCt^), also indicating the 95% confidence interval (CI).

### 4.4. Immunoblot Assay

We analyzed pro-CASP-3 and cleaved-CASP-3 protein levels using standard immunoblot procedures. We seeded 500,000 hippocampal neurons per well in 12-well plates with coverslips pre-coated with 50 µg/mL poly-L-lysine. Following this, 1 or 14 days later, we stimulated a set of wells overnight with 15 mM LGA. After 24 h, we incubated neuron lysates with radioimmunoprecipitation assay lysis buffer (RIPA, Sigma-Aldrich, St. Louis, MO, USA) containing a complete EDTA-free protease inhibitor cocktail (Roche) and centrifuged (14,000 *g* for 10 min at 4 °C). The protein content was determined using the bicinchoninic acid method (BCA protein assay kit, ThermoFisher Scientific). We ran and resolved a total of 50 µg of protein on a single SDS-polyacrylamide gel electrophoresis (SDS-PAGE) and then electrophoretically transferred it to a single 0.2 µm polyvinylidene difluoride membrane (PVDF; Immobilon, Merck Millipore, Burlington, MA, USA). Then, we cut the membrane into the necessary pieces for incubation and probing with the different antibodies: antibody against CASP-3 (rabbit anti-CASP-3, 1:1000; Cell Signalling Technology cat#9662, RRID: AB_10694681) and α-tubulin antibody as loading control (mouse anti-α-tubulin, 1:10,000; Sigma-Aldrich cat#T6074, RRID: AB_477582), according to the manufacturer’s protocol. Antibodies were functionally validated by the Cell Signalling Technology and Sigma-Aldrich companies, respectively. After incubation with the primary antibody, we washed the membranes with TBS-Tween20 (Sigma Aldrich) and incubated them for 1 h at RT with a horseradish peroxidase (HRP)-conjugated goat anti-rabbit secondary antibody (1:1000; Thermo Fisher Scientific cat#31460, RRID: AB_228341) or an HRP-conjugated goat anti-mouse secondary antibody (1:1000; Thermo Fisher Scientific cat#31430, RRID: AB_228307). Finally, we developed the HRP signal using the SuperSignal West Pico Chemiluminescent detection system (Pierce, Appleton, WI, USA) and measured it using ImageScanner III and LabScan v6.0 software (GE Healthcare BioSciences AB, Madrid, Spain) with the default settings.

### 4.5. Measurement of Caspase-3/7 Activity

To analyze the effector caspase activity in the stimulated hippocampal neurons, we employed the CellEvent™ caspase-3/7 green detection assay (ThermoFisher Scientific), which allows for analyzing activity in individual cells. Briefly, we seeded 35,000 hippocampal neurons per well in 24-well plates pre-coated with 10 µg/mL poly-L-lysine, and after 1 or 14 days, we stimulated cells O/N with 15 mM LGA. We assessed the effector caspase activity 24 h later by incubation in 2.5 μM of the assay reagent in warm PBS supplemented with 10% FBS medium for 30 min at 37 °C protected from light. We took photographs of the cells using an epifluorescence microscope (DMIL LED) with a 20× microscope lens, coupled to a Leica DFC 3000G camera. We used ImageJ software v.1.53c to process and analyze the images. We estimated the caspase-3/7 activity as the percentage of caspase-stained neurons related to the total number of neurons.

### 4.6. Calcein/Propidium Iodide Assay

We seeded 35,000 hippocampal neurons per well in 24-well plates pre-coated with 10 µg/mL poly-L-lysine. One or 14 days later, we stimulated cells O/N with 15 mM LGA before incubating them with 2.5 µM calcein-AM (Sigma-Aldrich, St. Louis, MO, USA) and 0.4 µg/mL PI (Sigma-Aldrich, St. Louis, MO, USA) in warm PBS supplemented with 10% FBS medium for 30 min at 37 °C protected from light. We took photographs of the cells using an epifluorescence microscope (DMIL LED, Leica Microsystem GmbH) with a 20× microscope lens, coupled to a Leica DFC 3000G camera. We used the ImageJ software v.1.53c to process and analyze the images. Calcein-AM labels viable cells, whereas PI gains access only to cells with plasma membrane damage and accumulates in the nucleus. We estimated neuronal survival or death from, respectively, the percentage of calcein-AM or PI-stained cells related to the total number of neurons.

### 4.7. Data Analysis

Data are expressed as the mean ± SD, as indicated in the figure legends. The normality of the data was verified using the Shapiro–Wilk test. If the data followed a normal distribution, the statistical significance of the treatment effects was tested using a two-tailed paired Student’s *t*-test or one-way or two-way ANOVA (followed by Tukey post-hoc test for pairwise comparisons), depending on the characteristics of the data. If the data did not follow a normal distribution, the statistical significance of the treatment effects was tested using a non-parametric Wilcoxon signed-rank test or a non-parametric Kruskal–Wallis test (followed by Dunn’s multiple comparison test for pairwise comparisons), depending on the characteristics of the data. The statistical analysis applied to each test is detailed in the figure legend, and the values of the normality analysis are reflected in the table in Appendix A. The statistical analyses were conducted and graphic representations drawn using Prism Software 5 (GraphPad Software Inc.) or R v3.4.3 (https://www.R-project.org/; accessed on 2 September 2022). Differences were considered statistically significant when the *p*-value was below or equal to 0.05.

All raw and processed data, plus the statistical analyses, are available as Appendix A at the Open Science Framework repository (OSF: https://osf.io/cusk8/?view_only=f2a4c1b866c14c9c8ed7c92ed8e0cb9a; accessed on 26 October 2023).

## 5. Conclusions

Our results point out that the observed opposite expression of endogenous Casp3 and miR-138-5p might influence the control of apoptosis and favor neuronal survival upon cell death stimulation at distinct neuronal maturation stages. However, further study would be needed to determine a more detailed role of miR-138-5p in apoptosis during neuronal maturation using loss-of-function assays with the downregulation of the microRNA using antagomiRs. Moreover, this regulation of apoptosis could be due not only to the action of miR-138-5p but also to the synergistic action of several microRNAs acting cooperatively on Casp3 or other targets in the apoptotic pathway. In conclusion, our results provide a broader role for miR-138-5p concerning neuronal maturation, regulating caspase 3 and favoring survival, to its already widely studied involvement in other pathologies, such as cancer.

## Figures and Tables

**Figure 1 ijms-24-16509-f001:**
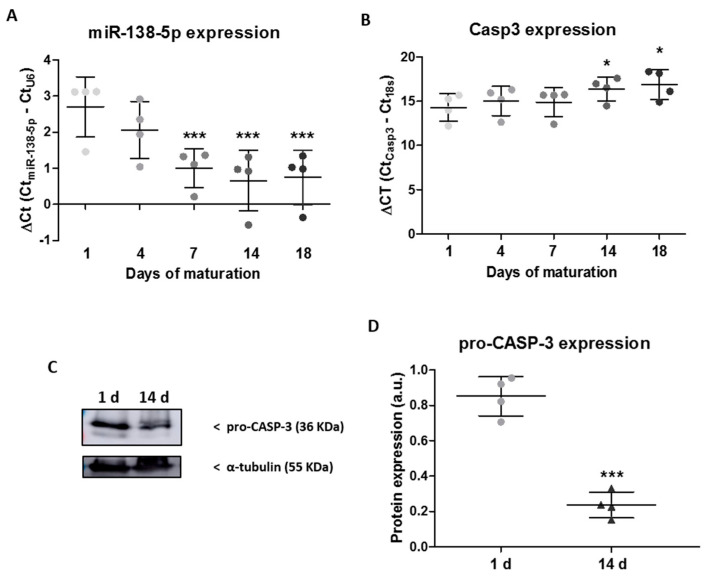
Endogenous expression levels of miR-138-5p and caspase 3 in hippocampal neurons over maturation time. (**A**,**B**) RT-qPCR showing the relative (**A**) miR-138-5p and (**B**) Casp3 expressions in RNA isolated from hippocampal neuron cultures at 1, 4, 7, 14, and 18 days of maturation. The expression of each gene (Ct) was normalized to the Ct of its corresponding control gene (snoRNA U6 for miR-138-5p (ΔCt = Ct_miR138-5p_ − Ct_U6_) and 18S for Casp3 (ΔCt = Ct_Casp3_ − Ct_18S_)). The dot plot represents the mean ± SD of four independent experiments. Statistical analysis was carried out using two-way ANOVA and Tukey post hoc tests. (**C**) Representative immunoblot images of the pro-CASP-3 and the load control α-tubulin protein expression in hippocampal neuron cultures at 1 or 14 days of maturation. (**D**) The dot plot summarizes the mean ± SD of the pro-CASP-3 protein expression of hippocampal neuron cultures at 1 or 14 days, normalized to the control protein expression α-tubulin of four independent experiments. Complete blot images are available at OSF (https://osf.io/cusk8/?view_only=f2a4c1b866c14c9c8ed7c92ed8e0cb9a; accessed on 26 October 2023). The statistical analysis was carried out with a two-tailed paired *t*-test. *, *** Denote significant differences at *p* < 0.05 and *p* < 0.001, respectively.

**Figure 2 ijms-24-16509-f002:**
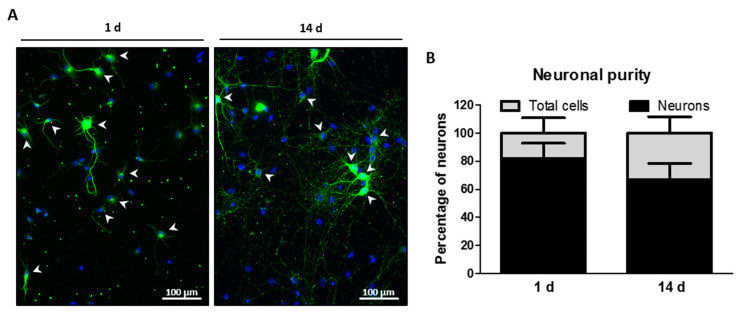
Purity of the hippocampal neuronal cultures. Cell purity in the hippocampal neurons at 1 or 14 days of maturation. (**A**) Representative epifluorescence images of hippocampal neuron cultures at 1 and 14 days, labeled with the specific neuronal marker β−III-tubulin (green) and DAPI (nuclei staining, blue). The white arrows show examples of labelled neurons. Bar scale = 100 μm. (**B**) The bar graph shows the mean of the percentage of neurons (β−III-tubulin-stained cells) related to total (DAPI positive) cells in culture ± SD of five independent experiments.

**Figure 3 ijms-24-16509-f003:**
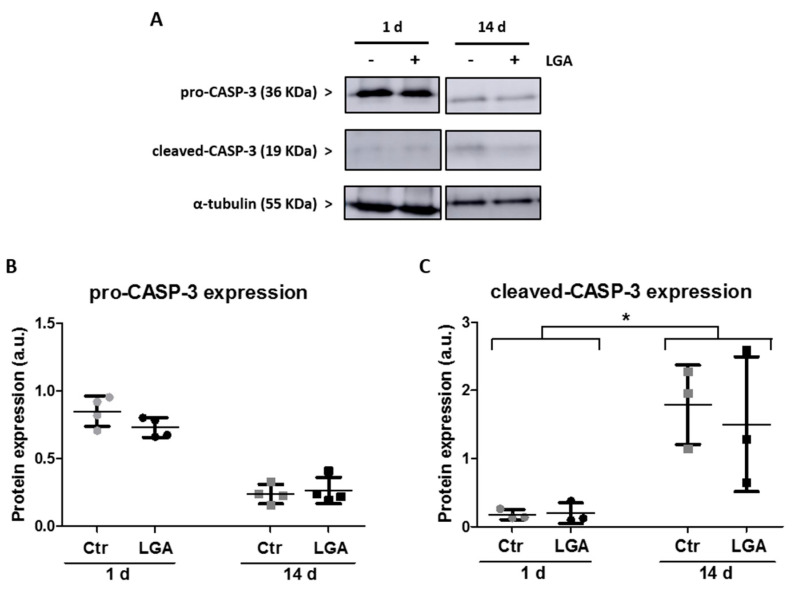
The expression of cleaved-CASP-3 was reduced with neuronal maturation. The protein expression of pro-CASP-3 and cleaved-CASP-3 was analyzed in hippocampal neuron cultures stimulated with LGA at 1 or 14 days of maturation. (**A**) Representative immunoblot images of pro-CASP-3, cleaved-CASP-3, and the load control α-tubulin protein expression in hippocampal neuron cultures treated with LGA or without treatment (Ctr) at 1 or 14 days. Complete blot images are available at OSF (https://osf.io/cusk8/?view_only=f2a4c1b866c14c9c8ed7c92ed8e0cb9a; accessed on 26 October 2023). (**B**,**C**) Dot plots summarize the mean ± SD of the pro-CASP-3 and cleaved-CASP-3 expressions of hippocampal neuron cultures at 1 or 14 days after LGA treatment, normalized to the control protein expression α-tubulin of three independent experiments. The statistical analysis of the ratio of cleaved-CASP-3 levels induced by LGA in neurons at 14 days related to the ratio of cleaved-CASP-3 levels induced by LGA in neurons at 1 day was carried out using a two-tailed paired *t*-test. * Denotes a significant difference (*p* < 0.05).

**Figure 4 ijms-24-16509-f004:**
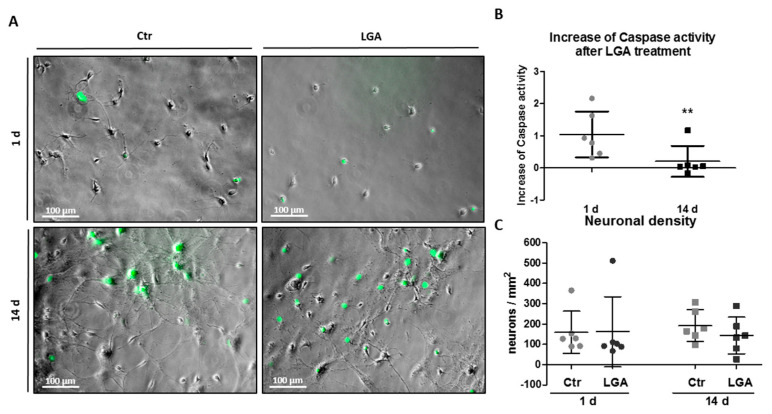
Neuronal maturation reduced the caspase activity. Caspase activity in hippocampal neurons after LGA stimulation at 1 or 14 days of maturation. (**A**) Representative phase contrast and epifluorescence images of hippocampal neuron cultures treated with LGA stimulus or without treatment (Ctr) at 1 or 14 days (green cells represent neurons stained with effector caspase reagent). Bar scale = 100 μm. (**B**) The dot plot represents the mean ± SD of the increase in the activated effector caspase-stained neurons after LGA treatment related to Ctr neurons (Increase of caspase-stained neurons (CNS) = (percentage of CNS cells after LGA−percentage of CNS in Ctr)/percentage of CNS in Ctr) of six independent experiments. The statistical analysis was carried out with a two-tailed paired *t*-test. (**C**) The dot plot summarizes the mean ± SD of the neuronal density (hippocampal neurons per mm^2^) of six independent experiments. The statistical analysis was carried out using a Kruskal–Wallis test and a Dunn’s multiple comparison post-hoc test. ** Denotes significant differences (*p* < 0.01).

**Figure 5 ijms-24-16509-f005:**
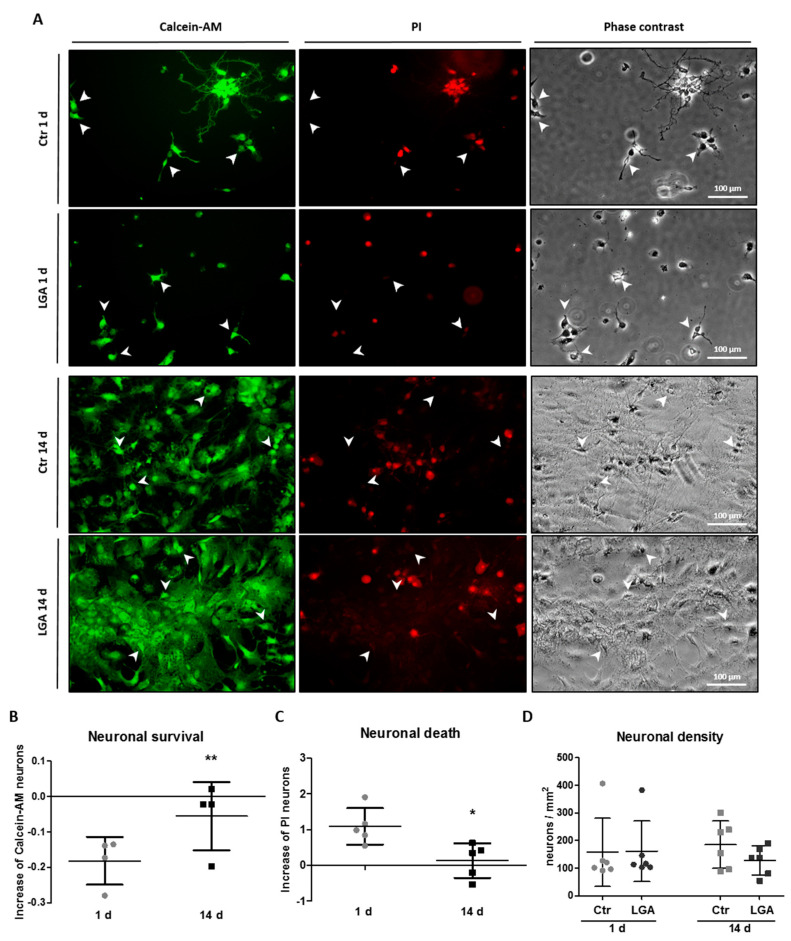
Neuronal maturation reduces cell death. Neuronal survival and death were analyzed in hippocampal neuron cultures stimulated with LGA at 1 or 14 days of maturation. (**A**) Representative phase contrast and epifluorescence images of hippocampal neuron cultures treated with LGA stimulus or without treatment (Ctr) at 1 or 14 days (images of green cells represent cells stained with calcein-AM; images of red cells represent PI-stained cells; images in phase-contrast represent total cells). The white arrows show examples of neurons stained with calcein-AM but not with PI. Bar scale = 100 μm. Dot plots summarize the mean ± SD of the change in the percentage of neurons stained with calcein-AM (surviving neurons) (**B**) and PI (dead neurons) (**C**) after LGA treatment related to Ctr neurons (Increase of calcein-AM neurons = (percentage of LGA cells − percentage of Ctr cells)/percentage of Ctr cells; Increase of PI neurons = (percentage of LGA cells − percentage of Ctr cells)/percentage of Ctr cells) of five independent experiments. The statistical analysis was carried out using a two-tailed paired *t*-test. (**D**) The dot plot represents the mean ± SD of the neuronal density (hippocampal neurons per mm^2^) of five independent experiments. The statistical analysis was carried out with a Kruskal–Wallis test and a Dunn’s multiple comparison post hoc test. *, ** Denote significant differences at *p* < 0.05 and *p* < 0.01, respectively.

**Figure 6 ijms-24-16509-f006:**
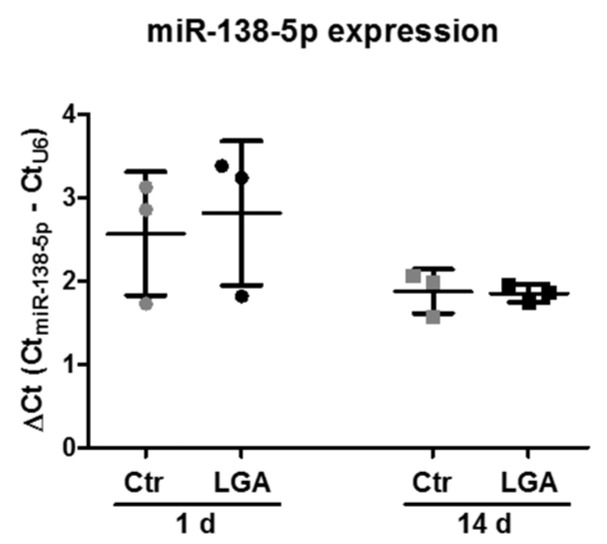
Differences in miR-138-5p expression after LGA stimulation at different days of maturation. RT-qPCR showing the relative miR-138-5p expression in RNA isolated from hippocampal neuron cultures stimulated with LGA at 1 or 14 days of maturation. The expression of miR-138-5p (Ct) from each sample was normalized to the Ct of the control gene snoRNA U6 (ΔCt = Ct_miR138-5p_ − Ct_U6_). The dot plot summarizes the mean ± SD of three independent experiments. The statistical analysis was carried out using a two-tailed paired *t*-test and a Wilcoxon signed-rank test.

## Data Availability

All information obtained while conducting the present analysis is available at Open Science Framework repository (OSF, https://osf.io/cusk8/?view_only=f2a4c1b866c14c9c8ed7c92ed8e0cb9a; accessed on 26 October 2023).

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
