# Peer review of "MiR-138-5p Upregulation during Neuronal Maturation Parallels with an Increase in Neuronal Survival"

_ijms, 2023, doi:10.3390/ijms242216509_

Round 1

Reviewer 1 Report

Comments and Suggestions for Authors

The manuscript brings interesting information concerning changes in miR-138-5p expression during neuronal maturation and the effect of this process on neuronal survival. The experiments were performed on hippocampal cell cultures and the expression of miR-138-5p and one of its targets – caspase-3. The manuscript is generally well-written and easy to read.

I have only a few minor comments:

1.       The authors used 10 mM LGA and showed no difference in neuronal density. Did they measure all cells or surviving cells? This is important, as such a concentration of LGA usually causes significant neuronal death (Fig. 4)

2.       2. Page 6, line 194 – “increased significantly less cell death..” – this phrase is hard to understand, please change it.

3.       Page 7, line 211 – figure legend, fig. B – “death neurons” – rather dead neurons.  Fig. B, C, vertical description – please indicate that presented values are  %

4.       Please place the section Methods after Conclusions

5.       It would be good if the authors tried to speculate how the results presented in the manuscript may relate to the developing brain.

Author Response

Comments and Suggestions for Authors

The manuscript brings interesting information concerning changes in miR-138-5p expression during neuronal maturation and the effect of this process on neuronal survival. The experiments were performed on hippocampal cell cultures and the expression of miR-138-5p and one of its targets – caspase-3. The manuscript is generally well-written and easy to read.

I have only a few minor comments:

  1. The authors used 10 mM LGA and showed no difference in neuronal density. Did they measure all cells or surviving cells? This is important, as such a concentration of LGA usually causes significant neuronal death (Fig. 4)

Answer: neuronal density is measured as the number of surviving neurons in the cell culture per well area. We use a concentration of LGA of 15 mM. We agree with the reviewer that this concentration of pro-apoptotic stimulus affects neurons, in fact it increases caspase activity with respect to control cells (unstimulated neurons). However, it does not significantly affect neuronal density at either 1 or 14 days of maturation, thus validating our results of caspase activity after LGA treatment.

Previously, we performed dose-response experiments in the laboratory using the MTT assay to see the effect of LGA on neuronal viability, obtaining a half-maximal inhibitory concentration (IC50) of 36.31 mM (see attached graph).

In the study of this manuscript we have used a concentration of 15 mM (less than half of the IC50) in order to cause an apoptotic effect on neurons (which causes some of them to die), but without being too high a concentration to avoid a significant death in cell culture.

  1. Page 6, line 194 – “increased significantly less cell death..” – this phrase is hard to understand, please change it.

Answer: we agree with the reviewer that this phrase is a bit complicated to understand. We have changed it to "reduced significantly cell death", leaving the phrase as "Complementary, hippocampal neuron cultures treated with LGA reduced significantly cell death at 14 days related to 1 day of culture (1 d = 1.09 ± 0.51, 14 d = 0.13 ± 0.48; two-tailed paired t-test, T4 = 4.227, p = 0.013; see Figure 5A and C)". Hopefully this will make it clearer that maturation reduces neuronal death following the pro-apoptotic LGA stimulus. Thank you very much for your suggestion.

  1. Page 7, line 211 – figure legend, fig. B – “death neurons” – rather dead neurons. Fig. B, C, vertical description – please indicate that presented values are  %

Answer: we agree with the referee. The correct phrase is “dead neurons”. We have corrected it in the manuscript.

The values represented in graphs B and C in figure 5 correspond to the increase in the percentage of surviving neurons (those in graph B, stained with calcein-AM) or dead neurons (those in graph C, stained with PI). These are not percentage values, but the change in that percentage at 1 or 14 days of maturation after LGA treatment related to control neurons (without stimulus). It is explained in the figure caption as “…change in the percentage of neurons stained with calcein-AM (surviving neurons) (B) and PI (death dead neurons) (C) after LGA treatment related to Ctr neurons (Increase of calcein-AM neurons = (percentage of LGA cells – percentage of Ctr cells) / percentage of Ctr cells; Increase of PI neurons = (percentage of LGA cells – percentage of Ctr cells) / percentage of Ctr cells)…".

  1. Please place the section Methods after Conclusions

Answer: thank you for the proposal. We have placed the Material and Methods section before the Conclusions section following the template and guidelines of the journal.

  1. It would be good if the authors tried to speculate how the results presented in the manuscript may relate to the developing brain.

Answer: because of the reviewer's proposal, we have suggested and speculated on how our results might relate to the development of CNS. To this purpose, and also to complement the comments of the other reviewer, we have added some comments in the discussion section (see in the track changes of the manuscript).

Reviewer 2 Report

Comments and Suggestions for Authors

In their paper entitled “MiR-138-5p upregulation during neuronal maturation parallels 2 with an increase in neuronal survival”, the Authors report that the opposite expression of miR-138-5p and its target Caspase3 might modulate apoptosis, a process recognized as a central determinant  of brain maturation during development.

Although, as recognized by the Authors, further work is required in order to understand how the observed regulation is obtained, the paper is of interest and suitable for Int. J. Mol. Sci. Moreover, Methods and Results are well described, and the figures are explicative.

 I should only propose to add, if possible, in the Discussion Section, a short proposal on the possible mechanisms regulating the expression of those miRNAs that can inhibit the apoptotic process in both developing and mature neurons.

Author Response

Comments and Suggestions for Authors

In their paper entitled “MiR-138-5p upregulation during neuronal maturation parallels 2 with an increase in neuronal survival”, the Authors report that the opposite expression of miR-138-5p and its target Caspase3 might modulate apoptosis, a process recognized as a central determinant  of brain maturation during development.

Although, as recognized by the Authors, further work is required in order to understand how the observed regulation is obtained, the paper is of interest and suitable for Int. J. Mol. Sci. Moreover, Methods and Results are well described, and the figures are explicative.

 I should only propose to add, if possible, in the Discussion Section, a short proposal on the possible mechanisms regulating the expression of those miRNAs that can inhibit the apoptotic process in both developing and mature neurons.

Answer: thank you very much for the proposal. In order to answer it, we have speculated on the possible mechanisms that regulate the expression of microRNAs that can inhibit the apoptotic process and, therefore, influence neuronal development and maturation. To this purpose, and also to complement the comments of the other reviewer, we have added some comments in the discussion section (see in the track changes of the manuscript).
